# Androgen-Independent Prostate Cancer Is Sensitive to CDC42-PAK7 Kinase Inhibition

**DOI:** 10.3390/biomedicines11010101

**Published:** 2022-12-30

**Authors:** Hyunho Han, Cheol Keun Park, Young-Deuk Choi, Nam Hoon Cho, Jongsoo Lee, Kang Su Cho

**Affiliations:** 1Department of Urology, Urological Science Institute, Yonsei University College of Medicine, Seoul 03722, Republic of Korea; 2Department of Pathology, Yonsei University College of Medicine, Seoul 03722, Republic of Korea; 3Pathology Center, Seegene Medical Foundation, Seoul 04805, Republic of Korea; 4Department of Urology, Prostate Cancer Center, Gangnam Severance Hospital, Yonsei University College of Medicine, Seoul 06229, Republic of Korea

**Keywords:** prostate cancer, androgen deprivation therapy, castration-resistant prostate cancer, staints, RNA-binding proteins

## Abstract

Prostate cancer is a common form of cancer in men, and androgen-deprivation therapy (ADT) is often used as a first-line treatment. However, some patients develop resistance to ADT, and their disease is called castration-resistant prostate cancer (CRPC). Identifying potential therapeutic targets for this aggressive subtype of prostate cancer is crucial. In this study, we show that statins can selectively inhibit the growth of these CRPC tumors that have lost their androgen receptor (AR) and have overexpressed the RNA-binding protein QKI. We found that the repression of microRNA-200 by QKI overexpression promotes the rise of AR-low mesenchymal-like CRPC cells. Using in silico drug/gene perturbation combined screening, we discovered that QKI-overexpressing cancer cells are selectively vulnerable to CDC42-PAK7 inhibition by statins. We also confirmed that PAK7 overexpression is present in prostate cancer that coexists with hyperlipidemia. Our results demonstrate a previously unseen mechanism of action for statins in these QKI-expressing AR-lost CRPCs. This may explain the clinical benefits of the drug and support the development of a biology-driven drug-repurposing clinical trial. This is an important finding that could help improve treatment options for patients with this aggressive form of prostate cancer.

## 1. Introduction

Prostate cancer is a common type of cancer among men. Androgen-deprivation therapy (ADT) is often used as a first-line treatment for recurrent or metastatic prostate adenocarcinoma, a type of prostate cancer where the androgen receptor (AR) drives tumor growth [1]. However, some patients develop resistance to ADT, and their disease is called castration-resistant prostate cancer (CRPC). This is often a lethal form of the disease.

To overcome ADT resistance, newer agents that target the androgen receptor have been developed, such as enzalutamide, apalutamide, and darolutamide [2,3,4,5,6]. However, the use of these agents has paradoxically led to an increase in non-AR-driven CRPCs [7,8]. One subtype of non-AR-driven CRPC is neuroendocrine prostate cancer (NEPC), which was once thought to be a rare subtype of the disease [9]. NEPCs express neuronal markers and are usually not sensitive to ADT. However, recent research has shown that NEPCs are more common than previously thought and can arise through the clonal expansion of neuroendocrine (NE) cells or through a state transition from adenocarcinoma [10,11]. NEPCs are characterized by loss of tumor suppressors *RB1* or *TP53* [12,13].

Another subtype of non-AR-driven CRPC is called double-negative prostate cancer (DNPC), which is negative for both the androgen receptor and classical NE markers. The biology of DNPC is not well understood, but it has been shown that FGF/MAPK signaling is active in these tumors [14]. It is possible that alterations at the epigenetic level, such as changes in microRNA expression, may play a role in the development of DNPC.

MicroRNAs (miRNAs) are small non-coding RNAs that play a key role in the regulation of gene expression through epigenetic mechanisms [15,16]. In cancer, miRNAs can act as either oncogenic drivers or tumor suppressors [17,18], and their expression profiles can be used to classify different types of tumors [17]. In prostate cancer, miRNA expression signatures have been shown to be useful for identifying and classifying tumors, and for understanding the biology of prostate stem and progenitor cells [18,19,20].

In this study, we used miRNA expression profiling to distinguish between different subtypes of prostate cancer, and found that miR-200 family under-expression was characteristic of DNPC. The common target of the miR-200 family, an RNA-binding protein called QKI, was found to be overexpressed in DNPC compared to other subtypes. Overexpression of QKI was associated with enhanced tumorigenicity and a transition to an androgen receptor (AR)-low state. Furthermore, QKI overexpressing cancer cells were selectively sensitive to the HMGCR inhibitor fluvastatin. Our findings provide new insights into the biology of DNPC and suggest potential therapeutic targets for this aggressive subtype of prostate cancer.

## 2. Materials and Methods

### 2.1. Cell Culture

LNCaP, MDA-PCa-2B and PC3 cell lines were purchased from ATCC. LNCaP and PC3 cells were maintained in RPMI-1640 (Gibco, Grand Island, NY, USA) supplemented with 10% FBS at 37 °C in 5% CO_2_. MDA-PCa-2B were maintained in BRFF-HPC1 (Athena Environmental Sciences, Baltimore, MD, USA) supplemented with 10% FBS at 37 °C in 5% CO_2_. Cells were fed twice weekly and split weekly with trypsinization.

### 2.2. RNA Isolation and MicroRNA Microarray

Total RNA was isolated using TRIzol reagent (Invitrogen, Carlsbad, CA, USA) according to the manufacturer’s protocol. The quality of RNA was confirmed using an Agilent 2100 Bioanalyzer (Agilent Technologies, Santa Clara, CA, USA). MicroRNA microarrays were performed using the Agilent Human miRNA v15 (AMDID 029297) per manufacturer’s instructions. The target prediction of miRNAs was surveyed using TargetScan database (release 7.1, [21]).

### 2.3. RNA-Sequencing Data Analysis

Cancer cell line RNA-sequencing dataset was acquired from Cancer Cell Line Encyclopedia (CCLE RNAseq gene expression data (RPKM), release date 14 February 2018). Metastatic prostate cancer dataset of SU2C/PCF Dream Team [22] and castration-resistant neuroendocrine carcinoma/adenocarcinoma dataset of Weil-Cornell Medicine [23] were downloaded from CbioPortal.org. Expression values of mRNAs and microRNAs in each dataset were transformed to z-score ((sample value−population mean)/sample standard deviation), which was used to generate gene set score. AR activity score and NE signature score were calculated as previously described [14].

PC3 cells treated with negative control siRNA or QKI-targeting siRNA (sequence: 5′-CUGGUAAUCGCCUUUGCUU UU-3′) [24]. Total RNA was extracted from samples using TRIzol (Invitrogen, Carlsbad, CA, USA) and 1 μg was sent to Macrogen Inc. (Seoul, Republic of Korea) for quality testing and library construction using TruSeq Stranded mRNA LT Sample Prep Kit (Illumina). Libraries were sequenced on a Illumina sequencer. RNA-seq reads were aligned to the human reference genome (GRCh37_NCBI_105) using HISAT2 version 2.1.0 (https://ccb.jhu.edu/software/hisat2/index.shtml (access date: 1 October 2022)).

Gene and transcript expression analysis was performed using StringTie version 2.1.3b (https://ccb.jhu.edu/software/stringtie/ (access date: 1 October 2022)). Differential expression analysis was performed by using the DESeq2 package from Bioconductor, considering genes with log2 fold change > 4 or <−2 and false discovery rate (FDR) < 0.05 as significantly differentially expressed. The output normalized count data from DESeq2 was used for downstream gene-set enrichment analysis (GSEA).

### 2.4. RT-PCR

To develop the RT-PCR blot, we first performed reverse transcription on total RNA samples using the cDNA synthesis kit (PB30.11 PCR Biosystems Ltd., Plymouth, PA, USA). The cDNA samples were then amplified using specific primers for the gene of interest and the PowerUp SYBR Green Master Mix (Thermo Fisher Scientific, Waltham, MA, USA) on a CFX96 Real-Time System (Bio-Rad, Hercules, CA, USA). Primers of QKI-5 and glyceraldehyde 3-phosphate dehydrogenase (GAPDH) sequences are described in the previous study [24]. The amplified cDNA was resolved on a 1.5% agarose gel and visualized using the GelDoc XR + system (Bio-Rad, Hercules, CA, USA). The blot was developed using the Enhanced Chemiluminescent (ECL) kit (Thermo Fisher Scientific, Waltham, MA, USA) according to the manufacturer’s protocol. The resulting image was captured using the ChemiDoc MP System (Bio-Rad, Hercules, CA, USA).

### 2.5. In Silico Drug Sensitivity and Genetic Perturbation Sensitivity Screening

We used the Cancer Dependency Map data of multiple cancer cell lines CRISPR screen (depmap.org) and the Cancer Therapeutics Response Portal data of a similar group of cell lines drug screen (https://portals.broadinstitute.org/ctrp/?page=#ctd2Target) (access date: 2 January 2020). CRISPR screen data are a matrix with the cell line identifiers as a row and each gene dependency score (CERES) as a column. A lower CERES score indicates a higher likelihood that the gene of interest is essential in a given cell line. Drug screen data are a matrix of cell line names as a column and each drug’s area under curve (AUC) value as a row. Therefore, the lower AUC, the more sensitive the cell is to the drug of interest. The two matrices were merged by using cell line names as identifiers. Then, a similarity matrix using Pearson correlation metric was generated. This new drug sensitivity—gene dependency score matrix contained all matched correlation coefficients for corresponding drugs and genes (481 drugs, 17,634 genes).

### 2.6. Target Validation in Cell Lines Using Western Blots

Equal amounts of cell extracts were subjected to SDS-PAGE and transferred to nitrocellulose membranes (Bio-Rad, Hercules, CA, USA). The following primary antibodies were used: anti-ZO-1 polyclonal (1:1000, Abcam, Cambridge, MA, USA), anti-E-cadherin (1:1000, Abcam, Cambridge, MA, USA), anti-Vimentin polyclonal (1:500, Abcam, Cambridge, MA, USA), anti-QKI (1:400; Abcam, Cambridge, MA, USA). Anti-rhoGDI (1:1000, Santa Cruz Biotechnology, Santa Cruz, CA, USA), anti-GAPDH (1:500, Santa Cruz Biotechnology, Santa Cruz, CA, USA) were used as internal control.

### 2.7. QKI-Overexpression by CRISPR/dCas9 System

QKI-overexpressing prostate cancer cell lines were generated by infecting QKI-lentiviral activation particles (Santa Cruz Biotechnology, Santa Cruz, CA, USA), using a synergistic activation mediator (SAM) transcription activation system designed to specifically and efficiently upregulate gene expression via lentiviral transduction of cells [25]. Briefly, cells were incubated with the target virus particle or control activation particle for 24 h with Polybrene. Infected cells were selected by puromycin for 96 hrs. Overexpression of the target gene was confirmed by qPCR and western blot.

### 2.8. In Vivo Tumor Xenograft and Drug Treatment

Tumor cells (1.5 × 10^6^ LNCaP cells/100 μL in 1:1 Matrigel: PBS solution; 1 × 10^6^ LNCaP cells/100 μL in 1:1 Matrigel:PBS solution;) were subcutaneously implanted into the left flank of nod-scid-gamma male mice (6–8 weeks old). Tumor size measurements were made weekly for PCa2B and biweekly for LNCaP with a caliper until the tumor volume became ≥1500 mm^3^. For drug treatment, animals were separated at 1-week post implant to vehicle control and PF-3785309 (Selleck Chemicals, Houston, TX, USA) treatment groups. PF-3758309 formulated in 0.5% methylcellulose was given 10 mg/kg twice daily via oral gavage.

### 2.9. Collection of Human Prostate Cancer Samples and Immunohistochemistry

We enrolled patients who received neoadjuvant bicalutamide therapy (150 mg/day) for 4 months before radical prostatectomy. Resected specimens were fixed in 4% buffered formalin and embedded in paraffin. Whole-mount step sections were cut axially at 5-mm intervals from the apex of the prostate to the tips of the seminal vesicle. After deparaffinization slides were routinely processed for immunohistochemistry (IHC) using the conventional avidin-biotin peroxidase complex technique. Anti-human chromogranin antibody (CHGA, Dako, Glostrup, Denmark) and anti-QKI c-terminal antibody (ab195960, Abcam, Cambridge, MA, USA) were used. Primary antibodies were diluted at 1:100 with Dako antibody diluents, with 0.05 M Tris-Hcl buffer containing 0.1% Tween to reduce background, and 15 mM sodium azide. Diaminobenzidine was used as a chromogen with hematoxylin counterstain.

### 2.10. RNA Interferences and Drug Treatment In Vitro

siRNA knockdown of genes CDC42 and PAK5 was performed by using Silencer^®^ Select pre-designed siRNAs (Thermo Fisher Scientific Inc.). Fluvastatin sodium hydrate (Sigma-Aldrich, St. Louis, MO, USA) was dissolved in water and further diluted with culture media. Vehicle control was generated by using equal amount of water.

### 2.11. Statistical Analyses

The relationships between two cell lines groups and the expression values obtained by microRNA array were analyzed using Welch’s ANOVA test. The associations between microRNA expression values and the other genes were analyzed by using Pearson correlation test. Results were considered to be significant at *p* < 0.05. Statistical analyses were computed with SPSS Statistics software (version 20, IBM, Chicago, IL, USA). All graphical presentations were created using GraphPad Prism software (version 6, GraphPad Software, San Diego, CA, USA).

## 3. Results

### 3.1. MicroRNA-200 Family Are Downregulated in Double-Negative Prostate Cancer (DNPC)

Previous transcriptomic analysis of metastatic castration-resistant prostate cancer (mCRPC) has identified gene sets that can classify different subtypes of prostate cancer, including AR positive prostate cancer (ARPC), NEPC, and DNPC (negative for both AR and NE markers) [14]. In this study, we applied this molecular subtype classification to the Weil Cornell Medicine (WCM) CRPC dataset of histologically confirmed adenocarcinoma (CRPC-Adeno) and NEPCs (CRPC-NE) [23]. Scatter plots of AR and NE signature scores showed three distinct clusters (Figure 1A). Most of the CRPC-NEs were classified as NEPCs, while a few CRPC-NEs and many CRPC-Adenos had negative (<0) AR/NE scores, which were classified as DNPCs (Figure 1A).

We then analyzed miRNA and mRNA expressions in the WCM dataset to identify distinct expression patterns among the molecular subtypes (Figure 1B, Appendix A). We found that 486 miRNAs (36.4%) were significantly differentially expressed between the subtypes (multiple *t* test, *p* < 0.05). We used the AR/NE scoring gene sets to redefine established prostate cancer cell lines into AR/DN/NEPCs (Figure 1C). A heatmap of the gene sets revealed three distinct groups: 22RV1, VCaP, LNCaP, and PCa2B as ARPC, PC3 and DU145 as DNPC, and NCI-H660 as NEPC. 22RV1 and VCaP expressed some NE markers, while DU145 expressed some AR markers. Therefore, we selected LNCaP and PCa2B as models of ARPC, and PC3 as a model of DNPC.

We compared miR expression levels in LNCaP and PC3 cells using microRNA arrays, and found that 6.4% of miRs were significantly overexpressed and 2.6% were significantly under-expressed in PC3 cells compared to LNCaP cells (Log2 fold difference > 2, *p* < 0.05). Notably, all five members of the miR-200 family (miR-141, -200a/b/c, -429) were among the top overexpressed miRs in LNCaP cells (Figure 1D). The miR-200 family has been shown to have a powerful influence on many steps of cancer progression, including metastasis and chemo-resistance [26], possibly reflecting their role in regulating epithelial to mesenchymal transition (EMT) [27]. Therefore, the positive enrichment of EMT and FGF/MAPK pathways in metastatic DNPCs may be related to repression of the miR-200 family [14]. Our next question was whether miR-200 family repression is specific to DNPCs in patients. We found that all five members of the miR-200 family were included in the list of 258 DNPC signature miRs (Figure 1B). We then generated an miR-200 family expression score by summing the z-scores of miR-200a,b,c, -141, and -429 in a sample. We found that DNPCs had the lowest miR-200 family expression levels among the three groups, while NEPCs had the highest expression (Figure 1E). This result provides indirect evidence that miR-200 family under-expression is specific to DNPCs.

### 3.2. QKI Is a MicroRNA-200 Family Target Gene That Is Overexpressed in Post-ADT Prostate Tumors

Since all five members of the miR-200 family were under-expressed in DNPC, we sought to identify a common target gene. We identified 218 putative target genes shared by the two broadly conserved miR-200b, -200c, -429 and miR-141, -200a subfamilies (Figure 2A). Using the human dataset as a filter, we narrowed down to 12 genes that showed inverse expressional correlations with the miR-200 family (Figure 2A). The top ranked (inversely correlated) gene was QKI, a homologue of mouse RNA-binding Quaking. QKI was also the only gene that was overexpressed in PC3 compared to other cell lines (Figure 2A). We confirmed QKI overexpression in PC3 by RT-PCR and Western blot (Figure 2B,C). We then compared the ability of the five miR-200 family members to inhibit QKI expression in PC3 cells, and found that miR-200b showed the most significant repression of QKI mRNA levels in PC3 (Figure 2D).

### 3.3. QKI Overexpression in Prostate Cancer Drives Tumor Progression and Epithelial-to-Mesenchymal Transition

In the previous study by Pillman et al., miR-200c was shown to target QKI in various types of epithelial cancers, which was linked to epithelial-to-mesenchymal transition (EMT) and metastatic progression [24]. Our study confirmed miR-200c’s ability to inhibit QKI mRNA expression (Figure 2D). Importantly, Pillman et al. also found that QKI was overexpressed in metastatic tumors compared to primary tumors, as well as in poorly differentiated tumors compared to well-differentiated primary tumors [24]. Furthermore, we observed that short-term ADT using charcoal-stripped serum media and enzalutamide increased QKI expression over time in LNCaP cells (Figure 2E). We also analyzed primary prostate tumor tissues that were exposed to long-term (6mo) ADT and bicalutamide treatment [10]. In these human samples, QKI expression levels were significantly higher in post-ADT tumors compared to pre-ADT tumors (Figure 2F and Appendix A). Overexpression of QKI in castration-resistant tumors was also validated in another prostate cancer dataset (Appendix A).

Based on our findings, we hypothesized that QKI may regulate the transition from AR-driven tumors to DNPC. To test this hypothesis, we generated QKI-overexpressing clones (QKI-OE) from LNCaP and PCa2B cell lines using a CRISPR/dCas9 activation system [25]. In vitro tumorsphere assays revealed an increase in the number and size of spheres formed by QKI-OE cells (Figure 3A and Appendix A).

Aldehyde dehydrogenase (ALDH) is a marker that has been used to identify cancer stem cells in various types of cancer, including prostate cancer [28,29]. Cancer stem cells are a subpopulation of cancer cells that have stem cell-like properties, such as the ability to self-renew and differentiate into different cell types. ALDH has been used as a marker to isolate and study these cells. We found that the ALDH-active population increased in both QKI-OE cell lines, suggesting characteristics of cancer stem cells (Figure 3B). In castrated host animals, the QKI-OE cells produced significantly larger palpable tumors than their parental cells (Figure 3C). Western blot analysis of EMT markers showed mixed characteristics of these cells, with a decrease in the tight junction protein ZO-1, no change in E-cadherin expression, and an increase in Vimentin expression (Figure 3D). Additionally, QKI-OE LNCaP cells showed heterogeneity in terms of cell morphology, with some cells exhibiting a flat and wide-spreading shape (Appendix A). This result is in line with previous work showing that QKI overexpression generally enriches mesenchymal characteristics [30]. However, it is likely that QKI itself does not affect the core program of epithelial characteristics, as represented by E-cadherin. Pillman et al. also reported that QKI siRNA knockdown in mesenchymal-like cancer cell lines did not affect *CDH1* expression [24].

### 3.4. QKI Knockdown Decreases Expressions of Cell Cycle and EMT Genes in Prostate Cancer Cell Line

Our RT-PCR and immunoblotting showed that QKI is overexpressed in androgen-independent PCa cell lines DU-145 and PC3 compared to LNCaP and PCa2B (Figure 2B,C). We selected the PC3 as a QKI-overexpressing DNPC model, performed siRNA knockdown of the QKI gene and analyzed the transcriptomic changes by RNA-sequencing. Importantly, gene-sets related to cell cycle and cell proliferation such as G2M checkpoints, mTOR signaling and mitotic spindle genes were all negatively enriched in QKI-knockdown PC3 cells (Figure 4A). Notably, the EMT gene-set was not significantly enriched in QKI knockdown cells. The top downregulated genes by QKI siRNA knockdown include *MYOD1*, *PPFIA4* and *MYOT1* (Figure 4B). *MYOD1* is a transcription factor that plays a key role in the differentiation of skeletal muscle tissue [31], while *PPFIA4* are genes that are involved in prostate and colon cancer progression [32,33]. *MYOT* is a gene that is involved in the development of the heart and skeletal muscles [34]. In contrast, the top upregulated genes include *KRT4*, *ALDH3A1*, *CSMD3* and *INAVA*. The classic mesenchymal marker gene *VIM* was downregulated, while *CDH1* and *TJP1* were not affected. Among the stem cell marker *ALDH* genes, *ALDH3A2* decreased, but *ALDH3A1* increased in QKI-knockdown PC3 cells (Appendix A).

### 3.5. Characterization of a Therapy-Resistant Mesenchymal-like State and the Impact of GPX4 Inhibitors and Statins in Tumor Cells

Earlier, Schreiber et al. molecularly characterized a therapy-resistant high-mesenchymal cell state across diverse tumor types and identified a distinct class of drugs targeting the phospholipid glutathione peroxidase (GPX4) as potential therapeutic targets [35,36,37,38]. We followed a similar in silico drug-mining approach using the QKI gene expression as input and found that the top hits included GPX4 inhibitors ML210 and ML162, as well as HMG CoA reductase (HMGCR) inhibitors fluvastatin and lovastatin (Figure 5A). Since GPX4 inhibitors are not yet available in vivo, we focused on the statins and found that QKI-overexpressing cells showed increased sensitivity to fluvastatin (Figure 5B). We also confirmed increases in reactive oxygen species production after fluvastatin treatment in QKI-overexpressing cells (Figure 5C). This suggests that the increased sensitivity of QKI-overexpressing mesenchymal-like cells to statins and GPX4 inhibitors may be related to intracellular lipid peroxide accumulation [39]. Additionally, their predicted sensitivity to GPX4 inhibitors implies that drug-induced mesenchymal state may be relevant to DNPC. To further investigate this, we generated a fluvastatin sensitivity signature gene set and created a heatmap with AR/NE scoring gene sets in a separate SU2C mCRPC dataset (Figure 5D) [22]. The results showed a clear negative correlation with AR activity score, suggesting that this recently-emerged AR-independent CRPC subtype may be a good candidate to test fluvastatin’s antitumor effect [14].

### 3.6. Fluvastatin Inhibits the Growth of AR-Lost Castration-Resistant Prostate Cancer Tumors by Targeting the QKI and CDC42 Pathways

To better understand the mechanism of fluvastatin’s activity, we generated a drug and genetic perturbation sensitivity correlation matrix by combining drug sensitivity data from the CTRP database and Avana CRISPR-Cas9 genome-scale knockout library data available at DepMap.org [40] (Figure 6A). This matrix implies that if “drug-of-question”-sensitive cell lines are selectively vulnerable to a series of gene knockouts, then the “drug-of-question” is likely to mainly target those genes or their pathways. For example, BRAF inhibitors and MEK1/2 inhibitors show strong correlations with gene knockouts of *MAPK1*, *BRAF*, and *SOX10* (Appendix A). GPX4 inhibitors RSL3, ML210, and ML162 also show strong correlations with GPX4 and selenocysteine production pathway genes. Using this drug-gene knock-out correlation matrix, we picked the top 0.1% genes (18 of 17,634 genes, all *p* < 0.0001) that were positively correlated with fluvastatin cytotoxicity. The top hit was *CDC42*, a rho GTPase that has already been implicated as a key downstream effector of statins’ antitumor activity (Figure 6B) [41]. We generated another drug-genetic perturbation sensitivity correlation matrix by combining drug sensitivity data and DepMap RNAi library. Here, when a single gene *QKI* mRNA level was used, *CDC42* was again the top hit (Figure 6C). We also applied an in-house fluvastatin sensitivity score, resulting in similar hits (Appendix A). QKI also showed significant correlations with fluvastatin sensitivity (Appendix A). Interestingly, HMGCR did not correlate with fluvastatin sensitivity (Figure 6B). Instead, *UBIAD1* was a second top hit correlation with fluvastatin cytotoxicity. This gene encodes a prenyl-transferase, which is the target of geranylgeraniol in the degradation of HMGCR enzymes [42]. *FGFR1* was also on the list, suggesting an association with the FGF/MAPK pathway, which is the main characteristic of DNPC [14]. *GPX4* was also on the list, supporting the characteristic dependency of mesenchymal-like cells on the lipid peroxidase reduction pathway [35].

Using the fluvastatin sensitivity score, we checked two canonical pathway databases (PID and BIOCARTA) and found enrichment of CDC42-related pathways (Figure 6D and Appendix A). We found that CDC42, RAC1, and P21-activated kinase 4 (PAK4) correlated positively with fluvastatin sensitivity. Furthermore, we checked the expression patterns of all PAK kinases (PAK1-6) and found that PAK5 (PAK7) correlated positively as well (Figure 6D). As inferred from their names, PAK family kinases are regulated by Rho GTPase family members Rho, rac1, and cdc42. In particular, PAK7 activity is most responsive to CDC42 [43]. RNA interference of CDC42 and PAK7 significantly affected QKI-OE cell proliferation more than control cells (Figure 6G,H). Additionally, pharmacologic inhibition of type II PAKs by PF-3758309 blocked LNCaP QKI-OE tumor growth in vivo (Figure 6I).

In another approach, we analyzed the transcriptomes of prostate cancer tissues from patients with metabolic syndrome—a coexistence of dyslipidemia, diabetes, and obesity. These groups of patients are important because statin use was associated with increased progression-free survival in those who took the drug to treat hyperlipidemia [44]. In other words, it is possible that the tumors of obese and hyperlipidemic patients are biologically distinct from the rest of the population [45,46]. Figure 6E shows the top 10 overexpressed transcripts in their tumor vs normal tissue analysis. Interestingly, PAK7 was one of them, which is in contrast with general population data showing that PAK7 is downregulated in tumor vs matched normal prostate tissue (TCGA and GTEx primary prostate tumor and matched normal, Appendix A). Nevertheless, PAK7 expression positively correlated with QKI expression in both tumor tissues and normal tissues (Appendix A).

## 4. Discussion

This study shows that (i) miR-200 family members are downregulated in double-negative prostate cancer (DNPC) while its target gene QKI is upregulated. (ii) QKI induces castration-resistance and promotes transition from epithelial-like ARPC to mesenchymal-like DNPC. (iii) QKI-overexpression DNPC models were selectively sensitive to statins via inhibition of the CDC42/PAK7 axis.

The regulation of epithelial-to-mesenchymal transition (EMT) is the key cellular function of the miR-200 family [47,48]. ZEB1/2 is a well-established EMT inducer and target of the miR-200 family [47,49], but increasing evidence suggests ZEB1/2-independent routes connecting miR-200 and EMT. Pillman et al. previously published a report connecting miR-200 to regulating epithelial plasticity via QKI [24]. They also stated QKI is enriched in metastatic prostate cancer vs primary tumor analysis. Our result is in line with their findings, apart from the fact that QKI was chosen by their inverse expressional correlation with miR-200 in the metastatic CRPC cohort. On top of it, we report that QKI is overexpressed in tumors following long term ADT, and its overexpression was sufficient to promote castration resistance. Still, further study is required to confirm whether the level of miR-200 is affected by ADT, and whether QKI-mediated cellular state transition can also promote metastasis.

Our repeated experiment showed that epithelial marker E-cadherin did not decrease at most whereas mesenchymal marker increased in QKI-OE cells. We suppose that the cells may have undergone hybrid EMT that exhibits both epithelial and mesenchymal characteristics. EMT is typically associated with the loss of epithelial markers such as E-cadherin and the acquisition of mesenchymal markers such as vimentin and N-cadherin, and hybrid EMT is thought to be an intermediate stage [50]. Some studies have suggested that hybrid EMT may be more common in cancer than full EMT, and that it may allow cancer cells to exhibit enhanced migration and invasiveness while still retaining some epithelial characteristics, such as the ability to proliferate and form colonies [51]. More research is needed to fully understand the role of hybrid EMT in cancer and its potential as a therapeutic target.

The LNCaP cells were originally derived from a lymph node metastasis of a human prostate cancer and are known to express AR. Previous studies have reported that they have several mutations of the AR genes, which makes them resistant to ADT, a common treatment for prostate cancer [14,52,53]. Because of this, the LNCaP cells have been considered androgen-independent and representative of a subtype of CRPC. However, in our study, we used the AR/NE signature score to redefine established prostate cancer cell lines into AR/DN/NEPC subtypes. This score is based on the expression levels of AR and NE marker neural-specific enolase. Based on this score, the LNCaP cells (as well as PCa2B cells) are considered to be part of the ARPC subtype because they express AR, and the AR is active in these cell lines. In contrast, the PC3 cells are considered to be part of the DNPC subtype because they do not express AR or NE markers.

DNPC is a demanding subset of prostate cancer because there are no established specific therapeutic options, and the population has gradually increased since the introduction of next generation AR signaling pathway inhibitors [14]. Since overexpression of QKI promotes the rise of DNPC, inhibition of its activity might be beneficial in this subset of patients. However, we were unable to directly target QKI, and its downstream effectors were unclear as well. Instead, we utilized existing cancer cell line data on their drug sensitivity to identify drug-gable signaling pathways active in QKI-overexpressing tumors. This approach has shown its efficacy by discovering the dependency of GPX4 in mesenchymal-like and/or drug-persistent cancer cells [39]. We improved this in silico target identification tactics by adding the genetic interference sensitivities (CRISPR knock-out and RNAi knock-down) data. We were able to internally validate this approach by checking several well characterized classes of drugs targeting specific pathways. For example, BRAF inhibitors and MAPK inhibitors showed the highest correlations with knock-out of *BRAF*, *MAPK1*, *SOX10* and *MITF* [54]. GPX4 inhibitors showed the highest correlations with GPX4 as well as selenocysteine metabolism genes (*SEPSECS*, *EEFSEC*, *SECISBP2*) [39,55]. In addition, IGF1R inhibitors showed highest correlations with *IGF1R*, *IRS2* and *FURIN* [56].

Our findings imply that type II PAK kinases are activated by the Rho GTPase CDC42 in a mesenchymal-like subset of tumors, which generally overexpress QKI. We propose that CDC42 is the key downstream effector of statins’ selective antitumor effect. It has been shown that statins are beneficial in preventing prostate cancer progression, especially during the ADT [57]. In this setting, QKI expression may serve as a good biomarker to select a patient subgroup to try fluvastatin combination with ADT. Similarly, in colon cancer, statin use was associated with improved patient survival [58]. QKI, as a representative marker of the tumor’s mesenchymal characters, may serve as a biomarker of statin sensitivity in colon cancer where the type II PAK inhibitors have shown their selectivity against mesenchymal-like subgroup [59].

## 5. Conclusions

In this study, we found that miR-200 family under-expression is characteristic of DNPC, a subtype of CRPC. The miR-200 family regulates QKI expression, and overexpression of QKI in DNPC was associated with enhanced tumorigenicity and a transition to an AR-low state. QKI overexpressing cancer cells were selectively sensitive to the HMGCR inhibitor fluvastatin, which suggests potential therapeutic targets for DNPC. We also observed that QKI is overexpressed in castration-resistant prostate cancers due to repression of the miR-200 family. QKI-overexpressing mesenchymal-like tumors were selectively vulnerable to fluvastatin, which inhibits CDC42/PAK5 activation. This suggests that inhibition of CDC42/PAK5 by statins may be the underlying mechanism of statins’ antitumor activity. Further studies are needed to confirm the potential clinical relevance of these findings and to explore the potential use of statins as a treatment for DNPC.

## Figures and Tables

**Figure 1 biomedicines-11-00101-f001:**
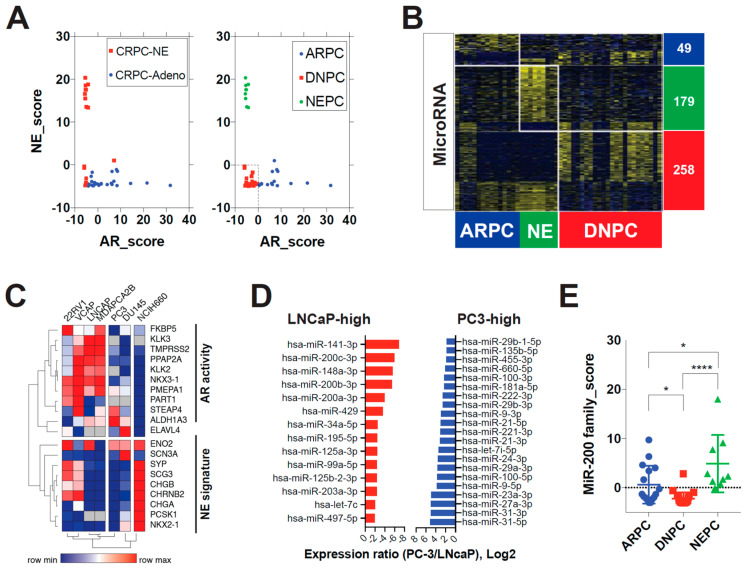
MiR-200 family expression is repressed in double-negative prostate cancer. (**A**) Molecular subtype classification applied to the Weil-Cornell-Medicine (WCM) CRPC dataset. Left: Histologic classification; Right: Gene expression-based classification. CRPC-NE = Castration-resistant prostate neuroendocrine carcinoma; CRPC-Adeno = Castration-resistant prostate adenocarcinoma. NE_score = mRNA expression z-score sums of 10 neuroendocrine signature genes. AR_score = mRNA expression z-score sums of 10 androgen receptor (AR) target genes. ARPC = AR-positive prostate cancer; DNPC = Double-negative (not expressing AR nor neuroendocrine markers) prostate cancer; NEPC = NE marker-positive prostate cancer. (**B**) Subtype-specific microRNA signatures for prostate cancer in the WCM dataset. Expression patterns of microRNAs according to molecular subtypes of prostate cancer (AR/NE/DN). Heatmaps for microRNA and mRNA show distinct expression patterns along with molecular subtype. Subtype-specific signatures were extracted by multiple *t*-tests between the subtype of interest and the other subtypes (Cut-off: *p* < 0.05). (**C**) Cell line model of DNPC. Hierarchical clustering by using AR activity and NE signature genes. Average Spearman rank clustering method was used. DNPC = double-negative prostate cancer, AR = androgen receptor; NE = neuroendocrine. (**D**) Top overexpressed miRs in LNCaP and PC3 cell lines. miR = microRNA. (**E**) MiR-200 family expressions in the WCM dataset. MiR-200 family_score = mRNA expression z-score sums of 5 miR-200 family genes. * *p* < 0.05; **** *p* < 0.0001.

**Figure 2 biomedicines-11-00101-f002:**
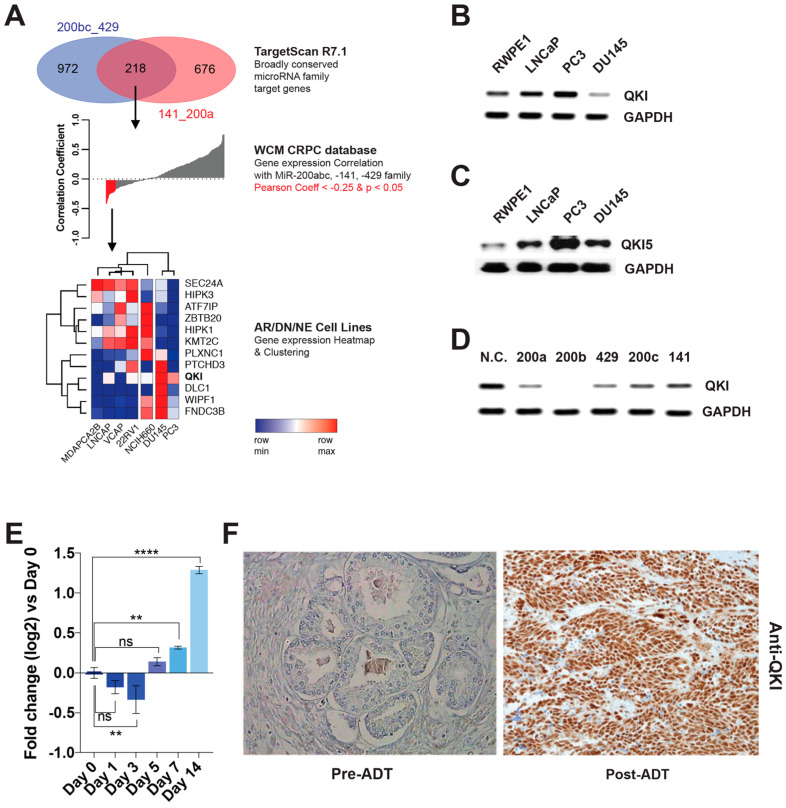
MiR-200 family target QKI is enriched in post-ADT prostate tumors. (**A**) MiR-200 familial target mRNA prediction by TargetScan. Two groups of putative target mRNAs by miR-200b, 200c, 429 and miR-141, 200a were combined, resulting in 218 genes that were overlapped. Their expressional correlation with the miR-200 family score in the WCM CRPC database was then calculated. Using −0.25 as a cut-off value, the expression of the selected 12 genes was analyzed in prostate cancer cell lines. Genes and cells were clustered to show their distributional relationship. In (**B**) RT-PCR and (**C**) western blot, PC3 overexpresses QKI compared to LNCaP and RWPE1 in. (**D**) QKI mRNA levels after treating the PC3 cell with miR-200 family mimics for 48 h. (**E**) QKI mRNA levels in LNCaP after culturing in charcoal-stripped serum media with 10 uM enzalutamide for 1–14 days. ns = not significant; ** *p* < 0.01; **** *p* < 0.001. Student *t*-test. (**F**) Immunohistochemistry staining of QKI in pre/post-androgen deprivation therapy (six month bicalutamide) exposed prostate tumor tissues (×200).

**Figure 3 biomedicines-11-00101-f003:**
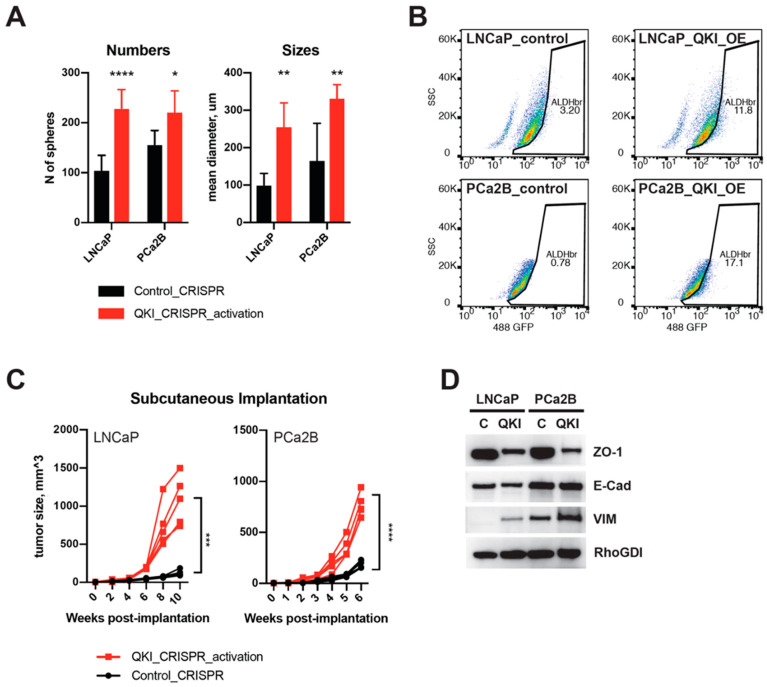
QKI promotes EMT and castration-resistant tumor growth. (**A**) Tumorsphere formation assays of QKI-overexpressing cells and control cells. Left: number of spheres; right: mean diameter of spheres. LNCaP and PCa2B cells with modified QKI expression were cultured in serum-free tumorsphere media in ultra-low attachment multiwell plates. Tumorspheres were measured after 10 d. (**B**) ALDH-active stem-like populations in QKI-overexpressing LNCaP (upper) and PCA2B (lower) cells. The ALDH<Br> population gate was set by DEAB (ALDH inhibitor) treated cells. The proportion of ALDH<Br> population was then measured in QKI-overexpressing cells and control cells. OE = overexpression. (**C**) Subcutaneous xenograft tumor formation assay. QKI-overexpressing cells and control cells were injected (1 × 10^6^ cells, 1:1 mixture of matrigel) subcutaneously into the flank of castrated NOD-SCID-gamma male mice. Tumor size was measured biweekly (LNCaP) or weekly (PCa2B) after implantation. (**D**) western blot of EMT markers in QKI-overexpressing cells. * *p* < 0.05; ** *p* < 0.01; *** *p* < 0.001; **** *p* < 0.0001.

**Figure 4 biomedicines-11-00101-f004:**
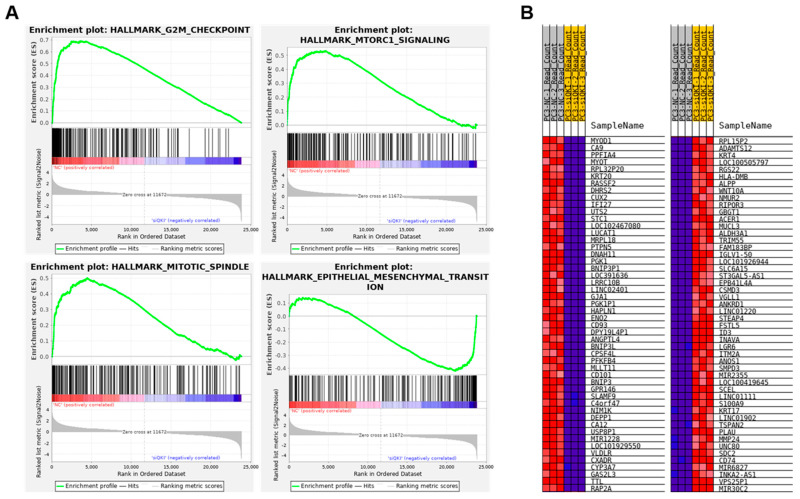
RNA-sequencing and gene-set enrichment analysis of QKI-knockdown PCa cell. (**A**) Enrichment plots of selected hallmark gene-sets. G2M_checkpoint, MTOR2_siganling and Mitotic_spindle gene-set enrichment plots show enrichement in NC (negative control) siRNA samples. Epithelial_mesenchymal_transition gene-set enrichment plot do not show significant enrichment. (**B**) Heatmap correlation plots of signature genes in PC3-NC sample vs PC3 siQKI (siRNA knockdown of QKI gene) sample. Top positive/negative genes are shown.

**Figure 5 biomedicines-11-00101-f005:**
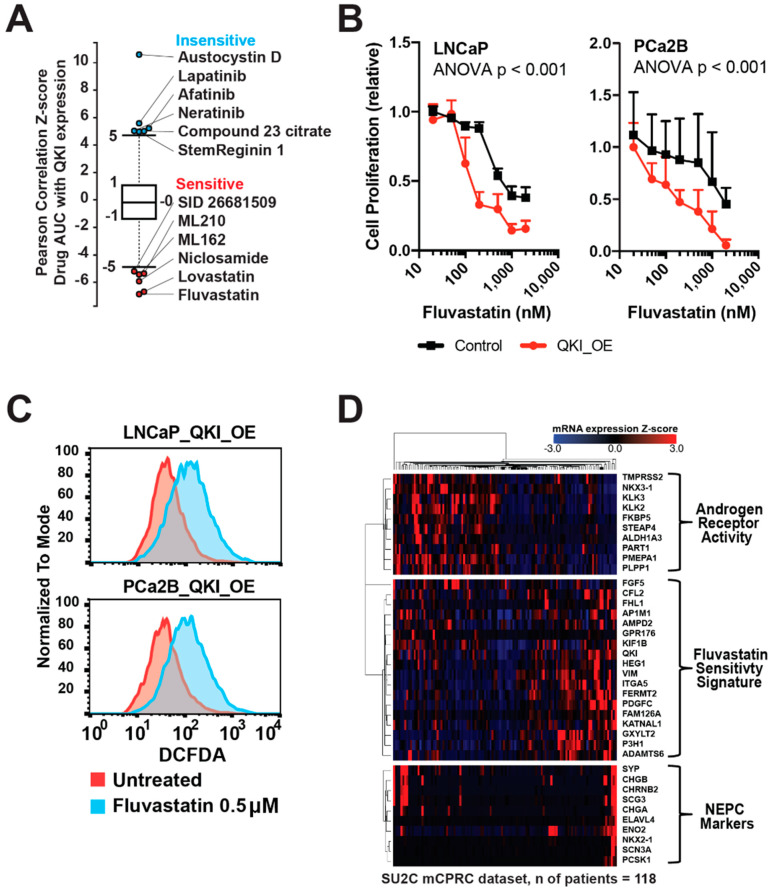
QKI-overexpressing CRPC cells are selectively sensitive to statins. (**A**) Correlation plot showing drug sensitivity (AUC) and QKI gene expression profile. *Y*-axis represents the correlation coefficient z-score for drugs (compounds). The top and bottom 5% of z-score items are depicted, where lower z-score corresponds to higher sensitivity. Data acquired from the Cancer Therapeutics Response Portal. AUC = area under curve. (**B**) QKI-overexpressing LNCaP and PCa2B cell proliferation in culture with fluvastatin after 5 d. (**C**) Reactive oxygen species formation measured by DCFDA fluorogenic dye intensity in cells exposed to fluvastatin. DCFDA = 2′,7′–dichlorofluorescin diacetate. (**D**) Fluvastatin sensitivity signature enrichment in a metastatic CRPC dataset (Robinson et al., Cell, 2015). SU2C = Stand Up To Cancer.

**Figure 6 biomedicines-11-00101-f006:**
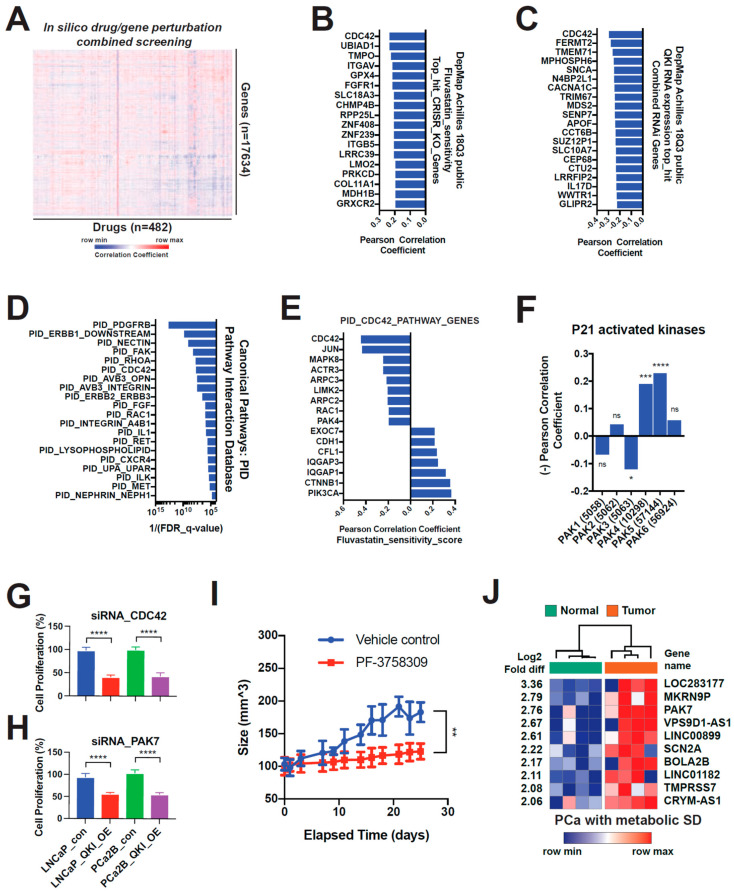
Inactivation of CDC42-PAK7 is a downstream mechanism of fluvastatin’s anti-tumor action. (**A**) Drug-genetic perturbation sensitivity correlation matrix. Pearson correlation coefficient was calculated across CTRP drug sensitivity data and the Avana CRISPR-Cas9 genome-scale knockout library data. Heatmap was generated by hierarchical clustering of the row (knockout genes) and the column (drugs). CRISPR knock-out library effect profile was acquired from DepMap.org (DepMap Public 18Q3 release). CTRP = Cancer Therapeutics Response Portal. (**B**) Top 0.1% genes with CRISPR knock-out sensitivity profiles that positively associate with fluvastatin sensitivity. (**C**) Top 0.1% genes with RNAi sensitivity profiles that positively associate with QKI mRNA level. Combined RNAi (Broad, Novartis and Marcotte) library effect profile was acquired from DepMap.org. (**D**) Top canonical pathway (PID, Pathway Interaction Database) gene-sets enriched in fluvastatin-sensitive cells vs. insensitive cells. Gene-sets were acquired from the Molecular Signatures Database (GSEA|MSigDB—Broad Institute). (**E**,**F**) Correlation profile of CDC42 pathway genes’ (**E**) and PAK family genes’ (**F**) RNAi sensitivity profiles to fluvastatin sensitivity score. (**G**,**H**) Cell proliferation upon siRNA knockdown of CDC42 (**G**) and PAK7 (**H**) compared to control siRNA in QKI-overexpressing LNCaP and PCa2B cells. Cell counting kit-8 (Dojindo Laboratories, Kumamoto, Japan) was used to measure relative proliferation rate in each condition. 450 nm absorbance was measured after 72 h. (**I**) Effect of type II PAK (PAK4, 5(7) and 6) specific inhibitor PF-3758309 on xenograft tumor growth. QKI-overexpressing LNCaP was implanted s.c. in castrated male mice and treated with vehicle or drug (n = 5 each). (**J**) Top overexpressed genes in prostate cancer tissue compared to matched normal glandular tissue in patients with metabolic syndrome (coexistence of hyperlipidemia, obesity, diabetes, and hypertension). * *p* < 0.05; ** *p* < 0.01; *** *p* < 0.001; **** *p* < 0.0001.

## Data Availability

All processed genomic, pathologic and clinical data are presented in Appendix A. Original genomic data will be provided upon request. The link and location of public data used in this paper is described in method section.

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
