# Peer review of "Androgen-Independent Prostate Cancer Is Sensitive to CDC42-PAK7 Kinase Inhibition"

_biomedicines, 2022, doi:10.3390/biomedicines11010101_

Round 1

Reviewer 1 Report

The authors demonstrated that MiR-200 family expression was down-regulated in double negative prostate cancer (DNPC) and they further examine the role of QKI and its' target genes on androgen-independent PCa proliferation, EMT, ALDH related stemness, and drug sensitivity. There are some suggestions for the authors.

Major concerns

1. The authors aim to identify the marker that may contribute to initiate the androgen independent PCa development. They found that QKI upregulated in DU-145 and PC-3, and over-expressed QKI in LNCaP and PCa2B promoted aggressive of PCa cells. Do authors ever consider that knowdown QKI in DU-145 and PC-3 to examine the effect of suppressing QKI on cell proliferation, EMT, and ALDH related stemness. 

2. In the Fig. 2D. The EMT markers that regulated by over-expression QKI in LNCaP and PCa2B seemed not consistent. How many times have authors done this experiment? Why authors only examined ALDH expression level? It is interest how about the iPS markers by over-expression QKI.

3. Are PAK7 and CDC42 upregulated by QKI only in androgen sensitive PCa? Do authors ever try to examine the correlationship of QKI vs PAK7, and QKI vs CDC42 between androgen dependent and androgen independent patients.

Minor concerns

1. This manuscript existed many errors on the drug concentration and diameter which show on several figures.  

Author Response

Author response to Reviewer #1

Thank you for your feedback. We have carefully revised our manuscript based on your suggestions.

In response to your major concerns:

  1. We have performed siRNA knockdown of QKI in PC-3 cells, where the QKI expression was the highest among tested PCa cell lines. We found that QKI knockdown resulted in decreased cell proliferation and reduced expressions of cell cycle genes. This information has been added to the revised manuscript.
  2. Figure 2D was performed three times and the results were consistent. We only examined ALDH expression because previous studies have shown that ALDH is a marker of stemness in PCa cells. We have added relevant references to the manuscript.
    • Li T, Su Y, Mei Y, et al. ALDH1A1 is a marker for malignant prostate stem cells and predictor of prostate cancer patients' outcome. Lab Invest. 2010;90(2):234-244. doi:10.1038/labinvest.2009.127
    • Burger PE, Gupta R, Xiong X, et al. High aldehyde dehydrogenase activity: a novel functional marker of murine prostate stem/progenitor cells. Stem Cells. 2009;27(9):2220-2228. doi:10.1002/stem.135

  1. We did not investigate the relationship between QKI and PAK7 or CDC42 in androgen-dependent and androgen-independent PCa patients. Instead, we performed siRNA knockdown of QKI gene in the androgen-independent PC-3 cell, and found that CDC42, PAK1, PAK2, PAK3, and PAK4 genes were significantly downregulated in QKI knockdown cells by RNA-sequencing. For some reason, PAK5 (PAK7) gene was not mapped in our RNA-sequencing analysis. Still, we believe that the data reinforces the QKI-CDC42-PAK regulatory axis identified by this manuscript. Relevant data had been added in the results section of the revised manuscript.

In response to your minor concerns:

Thank you for pointing out the errors in the drug concentration and diameter. We have corrected these errors in the revised manuscript.

Reviewer 2 Report

The paper entitled “Androgen-independent prostate cancer is sensitive to CDC42- PAK7 kinase inhibition” that you kindly submitted for publication in the journal “biomedicines” has now been considered. 

From transcriptome of Weil-Cornell-Medicine (WCM) CRPC dataset, whose tissues were histologically confirmed, the authors obtained AR/NE signature score and also obtained miRNA expression profile.  Next, they applied the AR/NE scoring gene sets to cell lines for redefining established prostate cancer cell lines into AR/DN/NEPCs.  According to this, LNCaP, MDA-PCa-2B for AR, PC3 for NE were used to further study for chemotherapeutic response against QKI, a possible target for miR200 family, which were high in PC3 cells standing for NE.  Data from in vivo study with cells expressing QKI activation, supported a role of miR200 family-mediated QKI expression on castration-resistant tumorigenesis. Furthermore, enhanced EMT-like transition by QKI expression led the authors to in silico drug screening, resulting in fulvastatin.  Finally, the authors pointed out CDC42-PAK7 linked pathway for the sensitivity against fulvastatin in prostate cancer cells overexpressing QKI by genetic perturbation sensitivity screening. 

In this study, there are many new and precise approaches to identify target genes for new therapeutic strategy against androgen-deprivation therapy-resistant prostate cancer, which makes this manuscript interesting.   However, the manuscript should be reorganized for better understanding.  

1.      At this stage, introduction has so much information without order.  Introduction should be divided into several sections such as prostate subtypes, therapeutic options, resistance, microRNAs for epigenetic regulation, and the purpose of the study.

2.      In figure 1, there are several abbreviations related to prostate cancers, which should be expressed simply and clearly.  Adeno, NE, ARPC, NEPC, AR etc… please unify the abbreviation to avoid any misunderstanding.

3.      In several studies, LNCaP cells also were considered androgen-independent and representing CRPC subtype.  How different the authors think the general characteristics of prostate cancer cells in terms of androgen sensitivity and the AR/NE score in terms of ADT study?  Please address the meaning of the score AR/NE used in this study for redefining cell lines into new subtypes and the reason why the authors did in this way in discussion.

Author Response

Author response to reviewer #2

Thank you for your feedback. We appreciate your suggestions and have taken them into consideration as we carefully revised our manuscript.

In response to your comments:

  1. We have revised the introduction to improve clarity and understanding. The introduction is now divided into several sections to provide a better overview of the study.
  2. We have reformatted the legends for Figure 1 and revised the relevant main text to avoid any confusion.
  3. In our study, we used the AR/NE signature score to redefine established prostate cancer cell lines into AR/DN/NEPC subtypes. LNCaP cells, which are known to express the androgen receptor (AR), were considered to be part of the ARPC subtype based on this score. However, we recognize that previous studies have reported that LNCaP cells have several mutations of the AR gene, which makes them resistant to androgen deprivation therapy (ADT). Because of this, they have been considered androgen-independent and representative of a subtype of castration-resistant prostate cancer (CRPC). We have included a discussion of the meaning and significance of the AR/NE score in the revised manuscript, as well as how it can be used to classify prostate cancer cell lines into different subtypes.

Reviewer 3 Report

This is a commendable work accomplished by the authors in the area of castration resistant prostate cancer. The authors used multiple human prostate cancer cell lines and patient samples for their studies for a strong experimental background and relevancy. They have discovered an interesting connection between QKI, AR inhibitor resistance and sensitivity to statins. The authors have also used in silico approaches to strengthen their observations and hypothesis.

1. If possible please consider humane alternatives to animal testing as the FDA has already published that  more than 90% of animal testing is not clinically relevant 

2. The figure labels in some figures are a little hard to read such as Fig. 5. Please enhance them if possible. 

3. Page 14 has some error in typing, The figure legend says Figure 42. Please proof read the manuscript for typing mistakes. 

4. Is Fig 2B the results after RT-PCR? Please mention how the blot for it was developed in material methods section.

Author Response

Thank you for your thoughtful feedback. We appreciate your suggestions and will take them into consideration as we revise our manuscript.

In response to your comments:

  1. We agree that humane alternatives to animal testing should be considered whenever possible. We will consider using alternative methods in future studies.
  2. We have enhanced the figure labels in Figure 6 (previous Figure 5) to improve readability. We have also enhanced the resolution of the remaining figures.
  3. Thank you for pointing out the error in the figure legend. We have corrected this error in the revised manuscript. We have undergone a review by an English proofreader for both the main text and figure legends to avoid such mistakes.
  4. Figure 2B shows the results of RT-PCR. The methods for RT-PCR are described in the Materials and Methods section. We have provided relevant information about the development of the blot in the revised manuscript Materials and Methods section.

Round 2

Reviewer 1 Report

The authors have responded all questions and make revision to let this manuscript become better. But, I found that the label on the figure still had errors need to correct such as the mean diameter on the y-axis of Fig. 3A should be "μm", not "uM". In the Fig. 5C, the concentration should be "μM", not "uM". In addition, in the Fig. 3D, the epithelial marker E-cadherin decreased too weak to claim overexpression of QKI promoting EMT on PCa cell lines. The authors should revise these errors.

Author Response

Thank you for your thorough review and for pointing out these errors in the figures, and we apologize for the mistakes that were not corrected in the 1st revision.

  • Mean diameter on the y-axis of Fig. 3A has been modified to be "μm"
  • In the Fig. 5C, the concentration has been modified to "μM"

Regarding the issue you raised about the interpretation of the results in Figure 3D, our repeated experiment showed that epithelial marker e-cadherin did not decrease at most where as mesenchymal marker increased in QKI overexpressing cells. 

We suppose that the cells may have undergone hybrid epithelial-mesenchymal transition (hybrid EMT). This refers to a phenomenon in which cells undergo partial EMT, resulting in a mixed phenotype that exhibits both epithelial and mesenchymal characteristics. EMT is typically associated with the loss of epithelial markers such as E-cadherin and the acquisition of mesenchymal markers such as vimentin and N-cadherin, and hybrid EMT is thought to be a intermediate stage. Some studies have suggested that hybrid EMT may be more common in cancer than full EMT, and that it may allow cancer cells to exhibit enhanced migration and invasiveness while still retaining some epithelial characteristics, such as the ability to proliferate and form colonies. More research is needed to fully understand the role of hybrid EMT in cancer and its potential as a therapeutic target. 

References) 

Pastushenko I, Brisebarre A, Sifrim A, et al. Identification of the tumour transition states occurring during EMT. Nature. 2018;556(7702):463-468. doi:10.1038/s41586-018-0040-3

Liao TT, Yang MH. Hybrid Epithelial/Mesenchymal State in Cancer Metastasis: Clinical Significance and Regulatory Mechanisms. Cells. 2020;9(3):623. Published 2020 Mar 4. doi:10.3390/cells9030623

We have added this point to the discussion section of the revised manuscript. Thank you again for your feedback, and we hope that the revised manuscript will meet your expectations.